# RETHINKING THE BERT-LIKE PRETRAINING FOR DNA SEQUENCES

## ABSTRACT

With the success of large-scale pretraining in NLP, there is an increasing trend of applying it to the domain of life sciences. In particular, pretraining methods based on DNA sequences have garnered growing attention due to their potential to capture generic information about genes. However, existing pretraining methods for DNA sequences largely rely on direct adoptions of BERT pretraining from NLP, lacking a comprehensive understanding and a specifically tailored approach. To address this research gap, we first conducted a series of exploratory experiments and gained several insightful observations: 1) In the fine-tuning phase of downstream tasks, when using K-mer overlapping tokenization instead of K-mer non-overlapping tokenization, both overlapping and non-overlapping pretraining weights show consistent performance improvement. 2) During the pre-training process, using K-mer overlapping tokenization quickly produces clear K-mer embeddings and reduces the loss to a very low level, while using K-mer non-overlapping tokenization results in less distinct embeddings and continuously decreases the loss. 3) Using overlapping tokenization causes the self-attention in the intermediate layers of pre-trained models to tend to overly focus on certain tokens, reflecting that these layers are not adequately optimized. In summary, overlapping tokenization can benefit the fine-tuning of downstream tasks but leads to inadequate pretraining with fast convergence. To unleash the pretraining potential, we introduce a novel approach called RandomMask, which gradually increases the task difficulty of BERT-like pretraining by continuously expanding its mask boundary, forcing the model to learn more knowledge. RandomMask is simple but effective, achieving top-tier performance across 26 datasets of 28 datasets spanning 7 downstream tasks. For example, RandomMask achieves a staggering 65.83% in Matthew's correlation coefficient for epigenetic mark prediction, which is a groundbreaking increase of 14.02% over the baseline and a remarkable 4.82% improvement over the SOTA results.

## 1 INTRODUCTION

In recent years, the combination of Transformer architectures, vast datasets, and unsupervised pretraining has led to a remarkable advancement in the field of natural language processing (Devlin et al., 2018; Floridi & Chiriatti, 2020; Zhao et al., 2023). Interestingly, a parallel can be observed between the semantic relationships in natural language and those found in DNA sequences. In natural language, words and sentences interrelate, while in DNA, elements such as promoters, enhancers, and transcription factor binding sites govern numerous biochemical interactions (Riethoven, 2010; Khoury & Gruss, 1983). The power of pre-trained language models lies in their ability to distinguish these subtle and interconnected relationships by pretraining on large-scale unlabeled data. Fortunately, gene sequencing projects, like the human genome project, have provided us with extensive collections of DNA sequence data (Gibbs, 2020) and paves the way for the development of DNA pre-training models. The prospect of utilizing pre-trained language model to uncover the hidden knowledge from vast DNA sequences is highly promising. Pioneering models like DNABERT (Ji et al., 2021), LOGO (Yang et al., 2022), and the Nucleotide Transformer (Dalla-Torre et al., 2023) demonstrate the early achievements in analyzing DNA sequences.

However, pre-trained models for DNA sequences often directly adopt methods from BERT (Devlin et al., 2018) in the field of NLP, neglecting the unique characteristics of DNA sequence. As shown in

Table 1: We present here the characteristics of relevant DNA downstream tasks, including Epigenetic Marks Prediction (EMP), Transcription Factor Prediction on the Human genome and the Mouse genome (TF-H and TF-M), Promoter Detection (PD), Core Promoter Detection (CPD), Splice Site Prediction (SSP), and Enhancer Activate Prediction (EAP). CLS and REG denote the classification and regression tasks, respectively.

| Downstream Tasks | EMP | TF-M | TF-H | PD | CPD | SSP | EAP |
|---|---|---|---|---|---|---|---|
| Task type | CLS | CLS | CLS | CLS | CLS | CLS | REG |
| Species | Yeast | Mouse | Human | Human | Human | Human | Drosophila |
| Sequence length | 500 | 100 | 100 | 300 | 70 | 400 | 250 |
| Single nucleotide | ✓ | ✓ | ✓ | ✓ | ✓ | ✓ | ✓ |
| Regional level | - | - | - | ✓ | ✓ | - | ✓ |

Figure 1: Comparison of Masked Language Models: From Natural Language Processing to DNA Sequence Analysis. In the experiments of this paper, both DNABERT and NT utilized 6-mer. For illustrative purposes, the figures use 3-mer as a representation.

Figure 1, there is a parallel between the masked language model used in both natural language processing and DNA sequence analysis. The figure highlights the divergent tokenization strategies employed for DNA sequences, with a focus on the K-mer tokenization method which adopts a sliding window approach, with 'K' symbolizing the window's size. Overlapping tokenization (DNABERT Ji et al. (2021)) occurs when the window shifts by a single step, while non-overlapping tokenization (the Nucleotide Transformer (NT) Dalla-Torre et al. (2023)) happens when the window moves by the size of 'K'. As in part 1(b), the NT, using non-overlapping K-mer tokenization, masks one token at the designated location and reconstructs it in the model's output. Conversely, part 1(c) illustrates DNABERT's approach, where under overlapping K-mer tokenization, 'k' consecutive tokens are masked at the chosen point and reconstructed in the final output. Unfortunately, these models still lack the analysis and consideration of the unique characteristics inherent in DNA.

DNA tasks differ significantly from tasks in natural language processing. In natural language, minor changes in letters generally have negligible impact. However, in DNA sequences, even slight variations at the nucleotide level can have a profound impact due to their high sensitivity (Benegas et al., 2022). For instance, sickle cell anemia (Kato et al., 2018) results from the substitution of the amino acid valine for glutamic acid at the sixth position in the $\beta$-globin chain (i.e., GAA changes to GTA), resulting in hemoglobin S replacing normal hemoglobin. Besides, some downstream tasks related to DNA sequences indeed focus closely on changes in individual nucleotides. An illustration is that Epigenetic Marks Prediction (Moore et al., 2013) zeroes in on pinpointing sites where specific biomolecules interact with nucleotides. On the other hand, certain DNA tasks underscore the model's proficiency in extracting broader, regional information. Promoter Detection, as demonstrated by Oubounyt et al. (2019), relies on the model's ability to effectively process sequences spanning tens to hundreds of base pairs. The diverse nature of these tasks is further illustrated in Table 1, highlighting the unique characteristics of each DNA downstream task.

To design pretraining methods tailored to the characteristics of DNA, we need a deeper and more comprehensive understanding of BERT-like pretraining methods for DNA. Specifically, we observed the following phenomena: 1) No matter whether the pretrained weights come from overlapping tokenization pre-training or non-overlapping pre-training models, adopting overlapping tokenization always improves performance when fine-tuning on downstream tasks. This might be because the overlapping tokenization method is sensitive to the changes in single nucleotide. 2) Overlapping techniques quickly produced distinct K-mer embeddings and exceptionally low loss, while non-overlapping methods resulted in vaguer embeddings and a slower loss reduction. 3) Using overlapping tokenization in pretraining, models have exhibited that in intermediate layers, self-attention

concentrates solely on particular tokens, showing a lack of diversity. This might indicate an under-training problem in these layers. In summary, although overlapping methods enhance fine-tuning performance, they can also introduce challenges during pretraining, such as rapid convergence and the risk of under-training.

Building upon the above observations, we further design for BERT-like DNA pretraining. While overlapping tokenization has proven beneficial in capturing intricate sequence patterns during the fine-tuning phase, particularly nuanced changes at the nucleotide level, it presents challenges during pre-training. Notably, models employing overlapping tokenization often exhibit rapid convergence, a phenomenon of "fast learning." This accelerated learning curve may inadvertently lead the model to overlook subtle interconnections within the data, culminating in a superficial pre-training process. Addressing this challenge, we introduce RandomMask. It is a technique grounded in the principle of progressively escalating task complexity for the model. By dynamically expanding masking boundaries throughout the BERT-like pretraining, RandomMask consistently presents the model with evolving challenges. This not only counteracts the model's predisposition to swiftly converge on solutions but also spurs sustained learning and adaptability. Empirically, our method achieves a new state-of-the-art on kinds of DNA downstream tasks.

Overall, the contributions of this paper can be summarized in the following three points:

1) We conducted a comprehensive exploration of BERT-like pretraining for DNA sequences, unveiling several interesting and significant observations. Our experiments emphasized that during the fine-tuning phase, employing K-mer overlapping tokenization consistently improves performance across both overlapping and non-overlapping pretrained weights. However, during the pretraining phase, common-used overlapping tokenization method leads to fast convergence and insufficient training.

2) To address these issues and unleash the potential of pretraining, we introduced Random-Mask, a novel approach that elevates the challenge in BERT-like pretraining by dynamically expanding mask boundaries, encouraging the model to assimilate richer knowledge.

3) We tested our approach on a total of 28 datasets spanning 7 downstream tasks. Across these tasks, we consistently achieved top-tier performance. Particularly noteworthy is our achievement in the widely-followed epigenetic mark prediction, where we secured a Matthew's correlation coefficient of 65.83%, surpassing the baseline by 14.02% and exceeding the current state-of-the-art by 4.82%.

## 2 RELATED WORK

### 2.1 BERT-LIKE DNA PRE-TRAINING APPROACHES

In recent years, significant advancements have been made in DNA pre-training, influenced by the success of BERT. DNABERT, introduced by Ji et al. (2021), applies BERT-like architectures to learn representations of DNA sequences. By leveraging Transformers' bidirectional nature, DNABERT captures dependencies and relationships between nucleotides, enabling a deeper understanding of genetic information (Le et al., 2021). It has demonstrated enhanced performance on tasks like DNA sequence classification, variant calling, and gene expression prediction. Another notable advancement is the Nucleotide Transformer (NT) proposed by Dalla-Torre et al. (2023). NT utilizes a significantly larger number of parameters compared to DNABERT, leading to notable performance enhancements. As the field continues to evolve, further refinements and novel approaches are expected, leading to more advanced analysis and interpretation of genetic information (Nguyen et al., 2023; Zhang et al., 2023).

### 2.2 K-MER TOKENIZATION

K-mer Tokenization involves dividing DNA sequences into subsequences or "k-mers" using a sliding window mechanism. Here, 'k' represents the window size, determining the length of each subsequence. Two commonly employed strategies within this framework are **Overlapping**, used by DNABERT, and **Non-overlapping** tokenization used by Nucleotide Transformer. To illustrate, let's consider the DNA sequence 'ATGACG' and tokenize it using a 3-mer approach. When using

Table 2: Performance comparison between overlapping and non-overlapping tokenization in the fine-tuning stage with two pre-trained models pre-trained with different tokenization methods. The results are reported in terms of MCC or PCC. The maximum values of MCC and PCC are both 100%. The larger these two indicators are, the better the model performance is.

| Model | Pre-trained |
|---|---|
| NT | Non-overlapping |
| DNABERT | Overlapping |

| Model | Fine-tuning | EMP | TF-M | TF-H | PD | CPD | SSP | EAP |
|---|---|---|---|---|---|---|---|---|
| NT | Non-overlapping | 45.37 | 39.81 | 55.25 | 87.71 | 62.56 | 80.39 | 38.94 |
| | Overlapping | **46.47** | **61.99** | **63.95** | **90.88** | **68.55** | **84.34** | **64.67** |
| DNABERT | Non-overlapping | 43.65 | 34.87 | 54.50 | 87.62 | 65.82 | 79.91 | 55.31 |
| | Overlapping | **51.81** | **59.60** | **63.55** | **90.48** | **72.84** | **85.44** | **68.43** |

overlapping, the resulting tokens are ATG, TGA, GAC, and ACG. On the other hand, when using non-overlapping, the tokens obtained are ATG and ACG.

## 2.3 SIGNIFICANCE OF SINGLE NUCLEOTIDE RESOLUTION

Single nucleotide resolution is crucial for a wide range of DNA-related tasks. Recognizing its significance, Nguyen et al. emphasized its importance in their study on HyenaDNA (Nguyen et al., 2023). They argued that the selection of an appropriate window size is crucial for models to accurately identify and extract information regarding individual nucleotides. Based on this observation, they advocated for a k-mer tokenization strategy, specifically employing a window size of one, to achieve enhanced resolution at the single nucleotide level.

## 3 EXPLORATORY EXPERIMENTS AND INSIGHTFUL OBSERVATIONS

To examine the effect of different tokenization methods, we performed two exploratory experiments and gain three insightful observations.

- It is common practice to adopt consistent tokenization methods for pre-training and fine-tuning. What if we use different tokenization methods for each stage? Conventional wisdom suggests that the performance may be negatively impacted because this inconsistency in tokenization hinders the model from leveraging the knowledge learned from pre-training. However, our findings demonstrate that overlapping tokenization consistently outperforms non-overlapping tokenization, regardless of the tokenization method pre-training employed. This suggests that, by its very nature, overlapping tokenization is superior for DNA downstream tasks.

- In order to delve deeper into the underlying differences between overlapping and non-overlapping tokenization, we conducted an extensive analysis of the pre-training process. This analysis allowed us to gain two more insightful observations: (1) Overlapping tokenization leads to a more organized embedding space with exceptionally reduced loss, while non-overlapping tokenization results in a chaotic embedding space with a slower but continuous decrease in loss. (2) The original MLM task is too easy for models using overlapping tokenization, thus hindering the sufficient training of attention mechanisms.

## 3.1 FINE-TUNING STAGE

> **Observation 1:**
> - During the fine-tuning stage, using overlapping tokenization instead of non-overlapping tokenization leads to consistent performance improvement for both overlapping (DNABERT) and non-overlaaping (Nucleotide Transformer) pre-trained models.

We performed a series of comparative experiments on diverse downstream benchmark tasks. Two pre-trained models were employed, namely "DNABERT" and "Nucleotide Transformer," both pre-trained on the whole human genome. "DNABERT" was pre-trained using overlapping tokenization, whereas "Nucleotide Transformer" was pre-trained using non-overlapping tokenization. Then we fine-tuned these two models on the benchmark consisting of seven downstream tasks with total 28 datasets. The results are shown in Table 2.

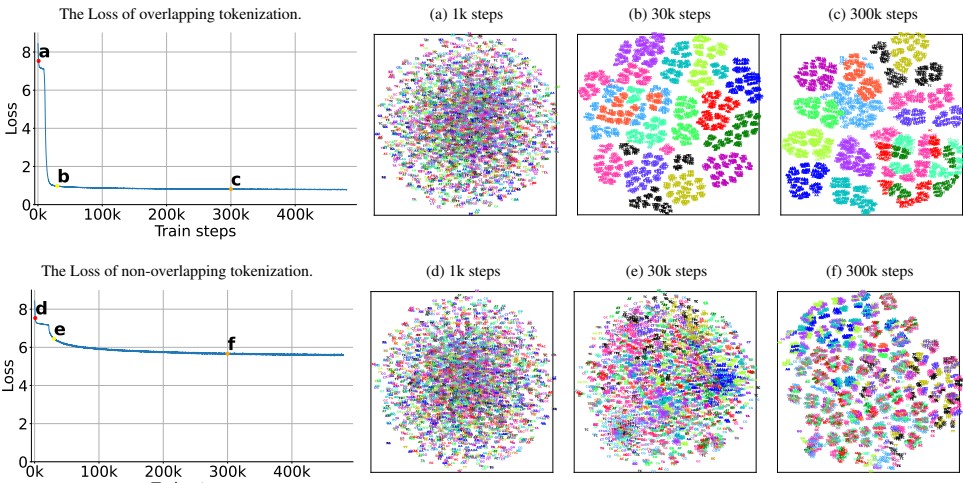

Figure 2: The loss curves with t-SNE visualizations of the embedding spaces during the training of DNABERT with overlapping tokenization (the first row) and DNABERT with non-overlapping tokenization (the second row). We observe that the magnitude of the loss value is inversely correlated with the level of organization observed in the embedding space.

In Table 2, we observe that regardless of the pre-training method employed, models fine-tuned with overlapping tokenization consistently outperform those utilizing non-overlapping tokenization. Specifically, DNABERT demonstrates improvements all 7 tasks, with an average increase of 10.25% in MCC or PCC. Similarly, Nucleotide Transformer also exhibits improvements in all 7 tasks, with an average increase of 10.01%.

We claim that the performance gap between overlapping and non-overlapping tokenization stems from the intrinsic superiority of overlapping tokenization for DNA downstream tasks. Addtionally, contrary to conventional belief, which suggests that inconsistency between pretraining and finetuning may hinder performance, our finding reveals that directly using overlapping tokenization leads to a significant improvement in performance of DNA downstream tasks, regardless of the chosen pre-training method.

## 3.2 PRE-TRAINING STAGE

To gain a deeper understanding, we perform an thorough analysis of the pre-training process. This involves pretraining two models, namely "DNABERT" with overlapping tokenization and "DNABERT" non-overlapping tokenization, on the entire Human Genome.

### 3.2.1 EMBEDDING SPACE ANALYSIS

> **Observation 2:**
> - During the pre-training stage, using overlapping tokenization results in a more organized embedding space, rapidly reducing the loss to an exceptionally low level. Conversely, using non-overlapping tokenization yields a more chaotic embedding space, with a continuous decrease in the loss.

We compare the the progression of embedding space along with loss values between the two models. To visualize the embedding space, we use the t-SNE algorithm (Van der Maaten & Hinton, 2008) and present the results in Figure 2. Comparing the two embedding spaces, we notice a notable distinction between the outcomes achieved by DNABERT when using overlapping and non-overlapping tokenization. For overlapping tokenization, as the loss decreases quickly, the embedding space becomes increasingly organized, resulting in a clear clustering of tokens when the loss reaches a low level. On the other hand, for non-overlapping tokenization, the loss continuosly decreases but remains at a relatively high level, with limited organization in the embedding space.

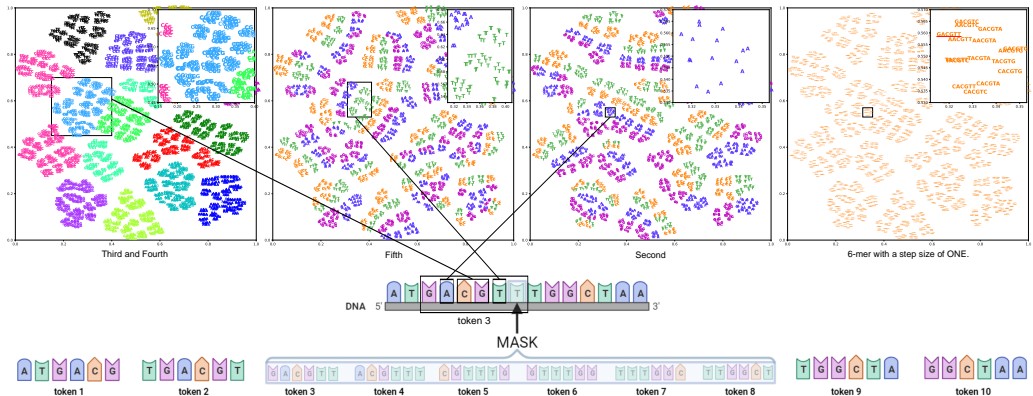

Figure 3: Detailed t-SNE visualization of the embedding space learned by DNABERT with overlapping tokenization. The first plot is the clustering of the central two nucleotides. The second and the third plot is the clustering of marginal nucleotides. The fourth plot is an illustration of the tokens in a specific cluster

Upon closer examination on Figure 3, we observe that each major cluster corresponds to the clustering of the central two nucleotides of each token and the distribution of tokens within the cluster is determined by the marginal nucleotides. We refer to these central two nucleotides in each token as the "representative elements" of the token. These representative elements establish the crucial one-to-one correspondence between tokens and nucleotides, which serves as the key factor contributing to the superior performance of overlapping tokenization.

We now give an intuitive analysis of the convergence of the two models. The rapid convergence and exceptionally low loss value of DNABERT with overlapping tokenization demonstrate the model's proficiency in solving the MLM task. However, it also implies that the pretraining task is too easy, leading to early overfitting. Nevertheless, The model's ability to recognize representative elements and utilizing the highly organized embedding space allow it to efficiently narrow down the search scope and accurately identify masked tokens. Consequently, the model effortlessly accomplishes the original MLM task, as masking six tokens is essentially equivalent to masking a single nucleotide, which is a relatively simple task.

### 3.2.2 ATTENTION ANALYSIS

> **Observation 3:**
> - Within the original MLM setting, overlapping tokenization leads to undertrained intermediate layers of the pretrained model, with the final layer focusing on a few nearby tokens. This indicates the existence of a shortcut, where the model heavily relies on the final layer to memorize mappings to the prediction.

As previously discussed, the rapid convergence and exceptionally low loss value of DNABERT with overlapping tokenization implies that the original MLM task is too simple for the model. This raises the possibility that the model has not been extensively trained, potentially limiting its ability to reach its full potential. In this section, we delve deeper into the analysis of the behavior of both models to validate the proposal and gain further insights

We visualize their attention machanism. The results are shown in Figure 4. We observe that the intermediate attention machanisms of DNABERT with overlapping tokenization are overly concentrated on the first token, namely the CLS token, with only the final layer focusing on a few nearby tokens. On the other hand, the attention machanism of DNABERT with non-overlapping tokenization are more evenly and diversely distributed across the sequence.

This phenomenon suggests that the model with overlapping tokenization effecly learns a shortcut, whereby it only relies on the final layer to memorize a limited set of mappings from nearby tokens to the output predicitons. Therefore, the intermediate layers remain mostly untrained. For model with non-overlapping tokenization, since the nearby tokens have no explicit information about the masked token, this shortcut is not available.

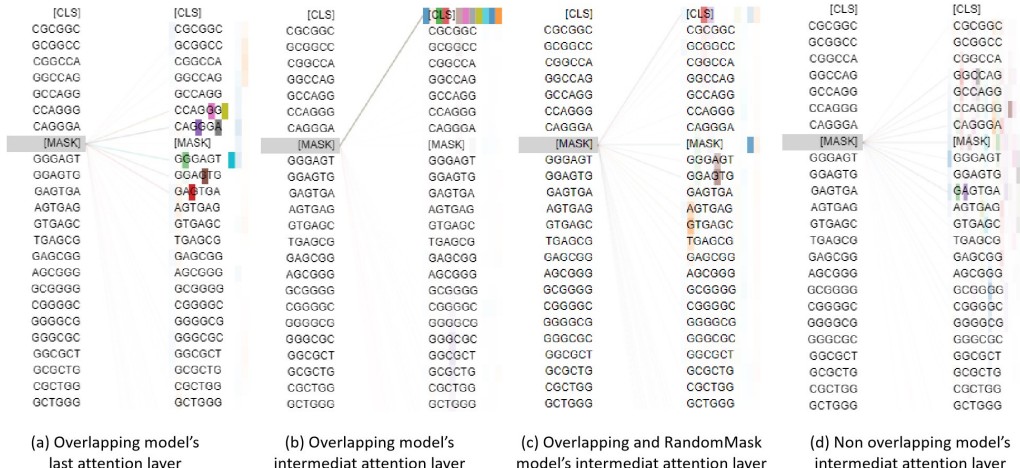

(a) Overlapping model's last attention layer

(b) Overlapping model's intermediat attention layer

(c) Overlapping and RandomMask model's intermediate attention layer

(d) Non overlapping model's intermediate attention layer

Figure 4: The attention mechanism of DNABERT with overlapping tokenization (a, b) and non-overlapping tokenization (d). We train DNABERT with RandomMask (c). The attention mechanism in intermediate layers mainly focuses on the first token, namely the CLS token, with only the final layer focusing on a few nearby tokens. RandomMask can addresses this issues. The attention machanism of DNABERT with non-overlapping tokenization exhibits a more diverse and evenly distribution across the sequence.

## 4 METHOD

Our method randomly expands the masking boundaries during the MLM pre-training stage, so we call it RandomMask.

**Tokenizer Method**: We adopt 6-mer overlapping tokenization for pretraining and funetuning, as delineated earlier, given its prowess in capturing a richer set of DNA sequence features. However, due to the accelerated convergence associated with the 6-mer overlapping tokenization during the pre-training phase, the model may experience insufficient training, thereby considerably constraining its performance. Consequently, we introduce a novel pre-training strategy.

**Pre-training Strategy**: To mitigate the drawbacks of overlapping tokenization during the pre-training phase, we need to progressively expand the masking boundary centered around the masking point, thereby pushing the model to continuously learn. Inspired by the curriculum learning strategy in Bengio et al. (2009), we segmented the 480k pre-training steps of DNABERT with 6-mer overlapping tokenization into five distinct phases. To ensure training stability, the length of consecutive mask tokens is randomly chosen between the minimum and maximum values. The minimum length of consecutive masks is set to 6, and the maximum length increases by increments of 2 at each stage. Specifically, in the training step $S$, the $MaskID$ of a DNA tokens sequence $X = (x_1, x_2, \ldots, x_n)$ are obtained through Algorithm 1, where $P$ is a pre-defined probability value, e.g., $P = 2.5\%$. Then, we can get mask tokens $\{x_i \mid i \subseteq MaskID\}$ for MLM pre-training.

---

**Algorithm 1** RandomMask

1: **procedure** RANDOMMASK($X, S, P$)
2:     Initialize empty set $MaskID$ and $masks \leftarrow [6]$
3:     Initialize $steps \leftarrow [30k, 60k, 100k, 150k, 480k]$
4:     **for** $i = 0$ **to** 3 **do**
5:         **if** $steps[i] < S \leq steps[i+1]$ **then**
6:             $masks \leftarrow [6, 8, \ldots, 6 + 2(i+1)]$
7:         **end if**
8:     **end for**
9:     $m \leftarrow$ uniformly select from $masks$
10:     **for** $i = 0$ to $len(X) - 1$ **do**
11:         Generate a real number $r \sim \mathcal{U}(0, 1)$
12:         **if** $r \leq P$ **then**
13:             $start \leftarrow i - m/2 + 1$
14:             $end \leftarrow i + m/2$
15:             **for** $i = start$ **to** $end$ **do**
16:                 **if** $0 \leq i \leq len(X) - 1$ **then**
17:                     Add $i$ to $MaskID$
18:                 **end if**
19:             **end for**
20:         **end if**
21:     **end for**
22:     **return** $(X, MaskID)$
23: **end procedure**

---

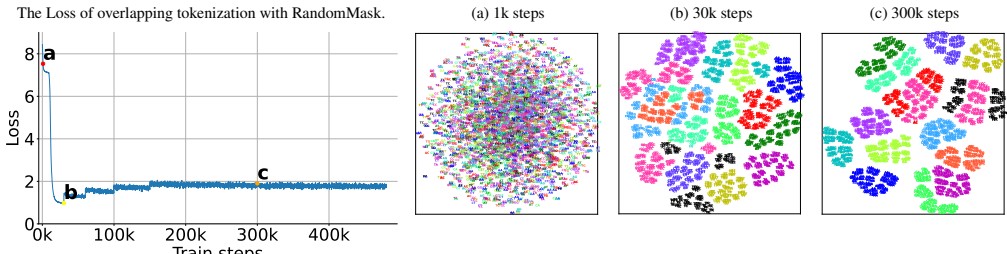

Figure 5: The loss curves with t-SNE visualizations of the embedding spaces during the training of DNABERT with overlapping tokenization and RandomMask.

**The loss curves and attention weight of RandomMask**: Figure 5 shows the training curve of overlapping DNABERT under RandomMask. It can be observed that after expanding the mask boundary in each phase, the loss value significantly increases, and the training curve for that phase shows a decline, indicating that the model is continuously learning. From the model's perspective, Figure 4(c) reveals that the intermediate layers of the model are trained with the use of RandomMask. Furthermore, we can witness a substantial performance boost in downstream tasks.

## 5 EXPERIMENTS

We train a Bert-like model on Human Genome (Gibbs, 2020) using the proposed RandomMask and evaluate it on seven downstream benchmark tasks consisting of 28 datasets in total. To ensure a fair comparison, all experiments are carried out using identical settings.

### 5.1 EXPERIMENTAL SETUP

**Architecture:** Our backbone network is designed following the configuration used in DNABERT (Ji et al., 2021), consiting of 12 Transformer Encoder layers with 768 hidden units and 12 self-attention heads. We adopt overlapping 6-mer tokenization method. The vocabulary size is 4,101, with 4,096 tokens representing the combinations of the four nucleotides in 6-mer arrangements and the remaining 5 tokens are reserved for special purposes.

**Pretraining:** For NT, hyenaDNA, and DNABERT, we used the pre-training weights available by the authors. For the DNABERT using RandomMask, we only modified the strategy for masking with RandomMask. Other settings, such as the model structure, training steps, and learning rate, remained consistent with DNABERT's settings.

**Finetuning:** We evalutate the models on seven downstream tasks consisting 28 datasets in total: Epigenetic Marks Prediction (EMP) (Pokholok et al., 2005; Phaml et al., 2005), Transcription Factor Prediction on the human genome and mouse genome (TF-H and TF-M), Promoter Detection (PD) (Oubounyt et al., 2019), Core Promoter Detection (CPD), Splice Site Prediction (SSP) (Wang et al., 2019), and Enhancer Activity Prediction (EAP). The majoriry of the data is from the GUE benchmark collected and proposed by Zhou et al. (2023), except for Enhancer Activity Prediction (EAP) from DeepSTARR (de Almeida et al., 2022). The hyperparameter setting for finetuning is adapted from Zhou et al. (2023) and Nguyen et al. (2023).

**Baseline:** For comparison, the following recent methods are chosen: DNABERT (Ji et al., 2021), Nucleotide Transformer (Dalla-Torre et al., 2023) and HyenaDNA (Nguyen et al., 2023). All models are trained on human genome and fine-tuned on the benchmark datasets with identical settings.

### 5.2 RESULTS

The results are shown in Table 3. Notably, RandomMask consistently outperforms the other methods on all seven benchmark tasks, achieving state-of-the-art performance in 26 out of 28 datasets.

For instance, when predicting epigenetic marks using sequence information alone, our method achieved an average Matthews Correlation Coefficient (MCC) of 65.83, surpassing the previous

Table 3: Performance of Different Methods on Seven Benchmark Downstream Tasks.

| Dataset | Epigenetic Marks Prediction (EMP) | | | | | | |
| --- | --- | --- | --- | --- | --- | --- | --- |
| | H3 | H3K14ac | H3K36me3 | H3K4me1 | H3K4me2 | H3K4me3 | H3K79me3 |
| NT | 69.67 | 33.55 | 44.14 | 37.15 | 30.87 | 24.06 | 58.53 |
| NT* | 72.60 | 39.11 | 44.25 | 35.47 | 27.59 | 23.49 | 59.14 |
| HyenaDNA | 77.57 | 61.80 | 59.71 | 49.82 | 44.86 | 58.17 | 65.74 |
| DNABERT | 75.82 | 48.07 | 51.52 | 43.92 | 31.01 | 37.13 | 58.98 |
| +RandomMask | **77.62** | **65.07** | **63.68** | **54.47** | **53.88** | **62.19** | **72.67** |

| Dataset | Epigenetic Marks Prediction (EMP) | | | | | |
| --- | --- | --- | --- | --- | --- | --- |
| | H3K9ac | H4 | H4ac | avg. | Enhancer | Splice Site |
| NT | 45.81 | 76.17 | 33.74 | 45.37 | 38.94 | 80.39 |
| NT* | 51.39 | 77.07 | 34.54 | 46.47 | 64.67 | 84.34 |
| HyenaDNA | 63.37 | 74.53 | 54.50 | 61.01 | 63.34 | 81.48 |
| DNABERT | 52.07 | 77.85 | 41.74 | 51.81 | 68.43 | 85.44 |
| +RandomMask | **65.02** | **79.44** | **64.22** | **65.83** | **69.56** | **87.20** |

| Dataset | Transcription Factor Prediction (Mouse) | | | | | | Core Promoter Detection | | |
| --- | --- | --- | --- | --- | --- | --- | --- | --- | --- |
| | 0 | 1 | 2 | 3 | 4 | avg. | notata | tata | all |
| NT | 31.04 | 75.04 | 64.67 | 29.17 | 29.27 | 45.84 | 66.58 | 71.91 | 63.01 |
| NT* | 50.54 | 77.73 | **78.05** | 61.01 | 42.64 | 61.99 | 68.71 | 73.90 | 68.55 |
| HyenaDNA | 47.55 | 79.85 | 74.58 | 58.77 | 41.81 | 60.51 | 63.77 | 64.16 | 63.97 |
| DNABERT | 46.27 | 78.84 | 74.41 | 59.04 | 43.45 | 60.40 | **71.88** | 76.06 | 70.47 |
| +RandomMask | **55.61** | **82.72** | 77.61 | **74.06** | **49.81** | **67.96** | 70.24 | **76.65** | **70.89** |

| Dataset | Transcription Factor Prediction (Human) | | | | | | Promoter Detection | | |
| --- | --- | --- | --- | --- | --- | --- | --- | --- | --- |
| | 0 | 1 | 2 | 3 | 4 | avg. | notata | tata | all |
| NT | 61.59 | 66.75 | 53.58 | 42.95 | 60.81 | 57.14 | 78.07 | **90.75** | 87.71 |
| NT* | 66.95 | 67.29 | 62.20 | 47.29 | 76.03 | 63.95 | 93.37 | 80.49 | 90.88 |
| HyenaDNA | 60.96 | 56.68 | 60.66 | 51.01 | 72.73 | 60.41 | 53.19 | 85.14 | 91.55 |
| DNABERT | 67.06 | 69.83 | 61.78 | 47.08 | 74.77 | 64.10 | 93.05 | 61.56 | 90.48 |
| +RandomMask | **67.13** | **72.55** | **71.64** | **60.14** | **77.20** | **69.73** | **93.40** | 84.03 | **92.74** |

best SOTA by 4.72% and DNABERT without RandomMask by 14.02%. In the case of enhancer activity prediction and splice site prediction, we reported a Pearson Correlation Coefficient (PCC) of 69.56% and an MCC of 87.20%, whereas the baseline achieved 68.43% and 85.44% respectively.

On transcription factor prediction, Regarding transcription factor prediction, our method demonstrated an improvement of 5.97% for mouse and 5.63% for human datasets. For promoter detection and core promoter detection, our approach outperformed other methods on 4 out of 6 datasets and achieved competitive performance on the remaining two datasets.

In conclusion, the application of the RandomMask strategy with overlapping tokenization significantly enhances the performance across various DNA downstream tasks.

## 6 CONCLUSION

While overlapping tokenization offers distinct advantages in fine-tuning for downstream tasks, its propensity for fast convergence can hinder comprehensive pretraining. RandomMask emerges as a potent solution, leveraging adaptive masking to push models to learn more effectively and deeply. By continuously increasing task difficulty and expanding mask boundaries, RandomMask ensures that models are well-equipped to handle both the nuances and broad patterns in DNA sequences.

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
