## A   APPENDIX

### A.1   DESCRIPTION OF METRICS

We introduce two important statistical metrics: the Matthews Correlation Coefficient (MCC) and the Pearson Correlation Coefficient (PCC). Both of these coefficients are crucial in assessing the relationship between variables in our model, with their maximum values reaching up to 100%. Higher values of these indicators signify better model performance.

1. **Matthews Correlation Coefficient (MCC):** The MCC is widely used in binary classification problems to evaluate the performance of such models. It is defined as:

$$MCC = \frac{TP \times TN - FP \times FN}{\sqrt{(TP + FP)(TP + FN)(TN + FP)(TN + FN)}}$$

   where:

   - TP = Number of True Positives
   - TN = Number of True Negatives
   - FP = Number of False Positives
   - FN = Number of False Negatives

   True Positives and True Negatives represent accurate predictions of the model, while False Positives and False Negatives denote incorrect predictions.

2. **Pearson Correlation Coefficient (PCC):** The PCC measures the strength and direction of a linear relationship between two continuous variables. It is calculated as:

$$PCC = \frac{\sum_{i=1}^{n}(\hat{y}_i - \bar{\hat{y}})(y_i - \bar{y})}{\sqrt{\sum_{i=1}^{n}(\hat{y}_i - \bar{\hat{y}})^2 \sum_{i=1}^{n}(y_i - \bar{y})^2}}$$

   where:

   - $\hat{y}_i$ = Predicted value by the model
   - $y_i$ = Actual value
   - $\bar{\hat{y}}$ = Mean of predicted values
   - $\bar{y}$ = Mean of actual values

For tasks such as EMP, TF-M, TF-H, PD, CPD, and SSP, which are classification tasks, the evaluation metric used is the MCC. In contrast, for the EAP task, which is a regression task, the evaluation metric is the PCC.

### A.2   COMPARISON OF RANDOMMASK METHOD WITH ALTERNATE TOKENIZATION APPROACHES

In addition to using the BPE tokenizer method, DNABERT-2 also changed the model architecture, adopted a new activation function, and utilized larger multi-species DNA data for pre-training. DNABERT2-BPE is an open source pre-trained weight from the author of DNABERT-2. To ensure a fair comparison, we trained DNABERT2-6mer and DNABERT2-6mer + RM using the 6mer tokenizer under the same architecture, pre-training data, and pre-training hyperparameters. The results showed in Table 1 in Supplementary Material. "+RM" indicates the use of the RandomMask strategy. It can be observed that the models using +RM show better performance. We plan to release the model weights and pre-training code open source soon.

### A.3   COMPARISON OF DNABERT MODELS ON ADDITIONAL BENCHMARKS

After comparison, some of the benchmarks used by HyenaDNA are already included in the GUE benchmark we used above. Below are some of the remaining datasets showed in Table 2 of Supplementary Material. We tested DNABERT1+RM. Since DNABERT2-6mer+RM has just completed pre-training, these downstream tasks are still running, and we will release them all later. The first three rows of results are from the original paper. The results in the original paper are accurate to one decimal place. For consistency, our test results in this table are only kept to one decimal place.

Table 1: Performance Comparison of Different Alternate Tokenization Models

| Model | EMP | TF-M | TF-H | PD | CPD | SSP | EAP |
|---|---|---|---|---|---|---|---|
| DNABERT | 51.81 | 60.40 | 64.10 | 90.48 | 71.88 | 85.44 | 68.43 |
| DNABERT + RM | 65.83 | 67.96 | 69.73 | **92.74** | 70.24 | 87.20 | 69.56 |
| DNABERT2-BPE | 64.47 | 68.00 | 70.11 | 87.91 | 70.53 | 84.99 | 67.79 |
| DNABERT2-6mer | 48.22 | 65.32 | 64.87 | 88.55 | 68.19 | 84.36 | 66.37 |
| DNABERT2-6mer + RM | **68.16** | **76.28** | **70.99** | 90.68 | **72.97** | **88.91** | **70.41** |

Table 2: Performance Comparison on Additional Benchmarks.

| | Avg. | Cod. vs Interg. | Hum. vs Worm | Hum. Enh. Cohn | Hum. Enh. Ens. | Hum. OCR Ens. | Hum. Reg. | Hum. Non-Prom. |
|---|---|---|---|---|---|---|---|---|
| CNN | 80.7 | 87.6 | 93.0 | 69.5 | 68.9 | 68.0 | 93.3 | 84.6 |
| Trans. | 84.4 | 88.8 | 95.6 | 70.5 | 83.5 | 73.0 | 91.5 | 87.7 |
| HyenaDNA | 88.3 | 87.6 | 96.5 | 73.8 | 89.2 | 80.9 | **93.8** | **96.6** |
| DNABETR | 87.8 | 93.2 | **97.0** | 74.3 | 89.0 | 81.0 | 88.5 | 91.7 |
| DNABERT + RM | **89.4** | **94.5** | 96.9 | **76.6** | **91.3** | **82.7** | 90.3 | 93.4 |

## A.4  COMPARISON WITH DIFFERENT LENGTHS OF K-MER

The mid-term Same-lenth involves copying the Non-overlapping tokens six times and splicing them together to obtain the same sequence length as Overlapping (i.e., Non-overlapping: token1 token2, Same-lenth: token1 token2 token1 token2 token1 token2 token1 token2 token1 token2 token1 token2).

Table 3: Expanded Comparison of Non-overlapping, Same-length, and Overlapping Tokenization Strategies for DNABERT

| Downstream Tasks | EMP | TF-M | TF-H | PD | CPD | SSP | EAP |
|---|---|---|---|---|---|---|---|
| Task type | CLS | CLS | CLS | CLS | CLS | CLS | REG |
| Sequence length | 500 | **100** | **100** | 300 | **70** | 400 | 250 |
| Same-lenth better | | ✓ | ✓ | | ✓ | | |
| **NT** | | | | | | | |
| Non-overlapping | 45.37 | 39.81 | 55.25 | 88.43 | 62.56 | 80.39 | 38.94 |
| Same-lenth | 44.88 | 47.59 | 60.57 | 86.96 | 63.98 | 80.96 | 37.98 |
| Overlapping | **46.47** | **61.99** | **63.95** | **90.88** | **68.55** | **84.34** | **64.67** |
| **DNABERT** | | | | | | | |
| Non-overlapping | 43.65 | 34.87 | 54.50 | 87.62 | 65.82 | 79.91 | 55.31 |
| Same-lenth | 42.98 | 38.60 | 53.27 | 85.33 | 64.09 | 80.76 | 38.62 |
| Overlapping | **51.81** | **59.60** | **63.55** | **91.76** | **72.84** | **85.44** | **68.43** |

An interesting phenomenon in Table 3 of Supplementary Material. In NT that uses overlap for pre-training, stretching the sequence length will indeed produce obvious gains in TF-M, TF-H and CPD. Combined with details of downstream tasks data, the common feature of these three tasks is that the DNA sequence length is short. The DNA sequence lengths of TF-M, TF-H and CPD are 100, 100 and 70 nucleotides respectively. The DNA sequence lengths of EMP, PD, SSP and EAP are 500, 300, 400 and 250 respectively.

But in general, the overlapping tokenizer to obtain more diverse tokens is better than simply lengthening the sequence length.

## A.5  ANALYZE REPRESENTATION OF THE MODELS

In the original DNABERT, the attention weight pattern is fixed. Adding RandomMask will diversify the attention weight pattern. In the Figure 1 and 4 of Supplementary Material, the comparison of the attention weights of different layers of DNABERT show that the attention weights of the 4th and 5th layers always focus on the [CLS] token. Taking a closer look, look at (a) in Figure 4 in Supplementary Material. We can see that adding RandomMask will diversify the attention weight

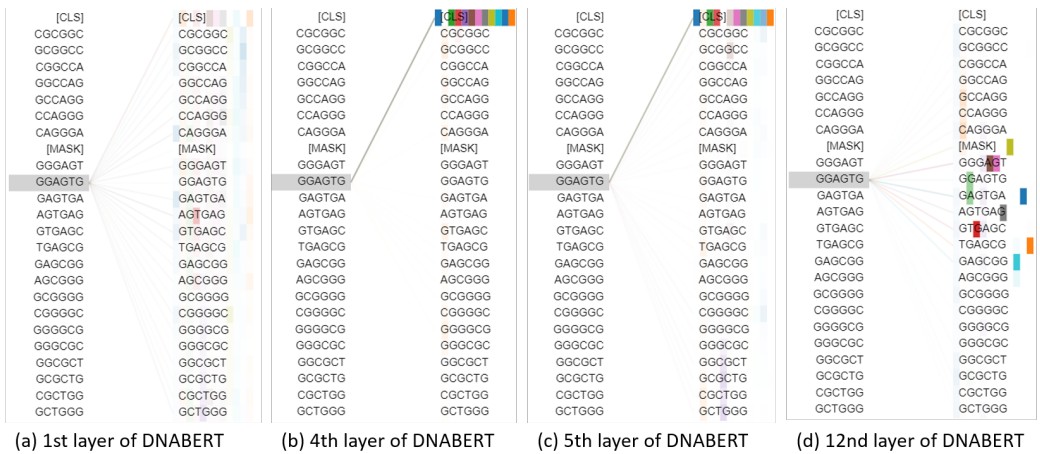

Figure 1: Attention weights from different layers of DNABERT.

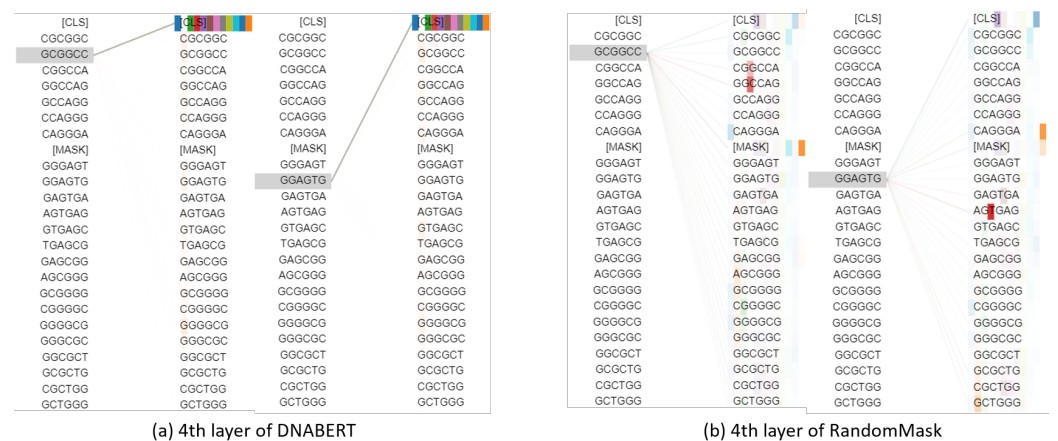

Figure 2: Attention weights from 4th layer of the different tokens. Compare the attention weight of different tokens DNABERT and RandomMask.

pattern. This is due to the fact that the model with RandomMask can continuously learn compared to the original model.

In Figure 3 of Supplementary Material, It can be seen that after adding RandomMask, the training curve can continue to decline. We believe that it is the pattern fixation of attention weights rather than the sparseness of attention weights that limits model capabilities.

## A.6 CURRICULUM LEARNING AND RANDOMMASK

Curriculum learning was proposed by Yoshua Bengio in 2009. It is a training strategy that mimics the human learning process by presenting the examples in an easy-to-difficult order. It has been shown to improve the performance and convergence stability of various large language models.

The paper in ACL2020 that explores curriculum learning for large language models, titled *Curriculum Learning for Natural Language Understanding*. This paper proposes a curriculum learning approach that reviews the training set in a crossed way to distinguish easy and hard examples and arranges a curriculum for large language models on various NLU tasks. We wanted to increase the difficulty of the DNA sequence pre-training task and at the same time ensure the stability of the training, so we adopted a curriculum learning solution.

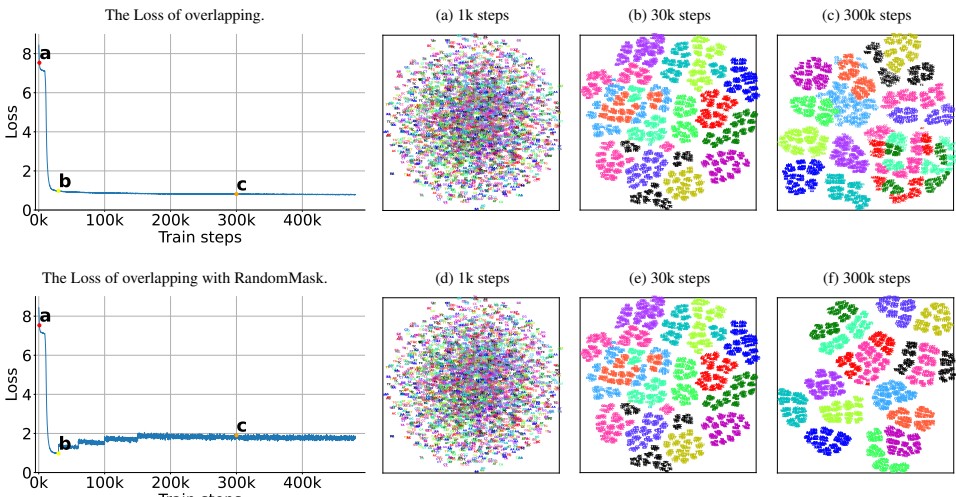

Figure 3: The loss curves with t-SNE visualizations of the embedding spaces during the training of DNABERT (the first row) and DNABERT with RandomMask (the second row).

In Table 4 of Supplementary Material, we tried consecutive masks 6, 12, 14, 28, and 40 tokens during the entire pre-training process. Finally, it was found that this RandomMask from mask 6 to mask 14 achieved the best performance.

Table 4: Performance Results with Different Continuously Mask Number and RandomMask.

| mask | EMP | TF-M | TF-H | PD | CPD | SSP | EAP |
|---|---|---|---|---|---|---|---|
| Mask 6 (DNABERT) | 51.81 | 60.40 | 64.10 | 93.05 | **71.88** | 85.44 | 68.43 |
| Mask 12 | 52.96 | 67.12 | 67.05 | 92.01 | 71.05 | 86.62 | 67.03 |
| Mask 14 | 51.44 | 66.86 | 66.60 | 92.59 | 70.67 | 85.31 | 69.16 |
| Mask 28 | 50.95 | 65.94 | 67.75 | 91.03 | 70.70 | 83.33 | 66.85 |
| Mask 40 | 49.93 | 66.19 | 67.54 | 90.21 | 70.01 | 82.35 | 63.37 |
| RandonMask | **65.83** | **67.96** | **69.73** | **93.40** | 70.24 | **87.20** | **69.56** |

In Table 5 of Supplementary Material, we tested both options from easy to hard and from hard to easy. In the table below, we divide pre-training into two stages equally. Mask 6 to 12 means masking 6 tokens continuously in the first stage and masking 12 tokens continuously in the second stage. Mask 12 to 6 means masking 12 tokens continuously in the first stage and masking 6 tokens continuously in the second stage. It can be found that the former has better performance.

Table 5: Performance results with different pre-training strategies

| strategy | EMP | TF-M | TF-H | PD | CPD | SSP | EAP |
|---|---|---|---|---|---|---|---|
| Mask 6 to 12 | **59.22** | **67.83** | **66.37** | **92.81** | 71.51 | **86.20** | **67.38** |
| Mask 12 to 6 | 53.15 | 63.47 | 65.18 | 90.42 | **71.93** | 83.95 | 66.37 |

## A.7 DETAILS IN FINETUNING

Table 6: Default hyperparameter settings for DNABERT and DNABERT + RandomMask in downsteam tasks.

| | EMP | TF | CPD | PD | SSP | EAP |
|---|---|---|---|---|---|---|
| Optimizer | | | AdamW | | | |
| Optimizer momentum | | | $\beta_1, \beta_2 = 0.9, 0.999$ | | | |
| Batch size | 32 | 32 | 32 | 32 | 32 | 64 |
| Training epoch | 100 | 10 | 10 | 5 | 10 | 10 |
| Learning rate | | | 3e-5 | | | |
| Weight decay | | | 0 | | | |

Table 7: Default hyperparameter settings for the Nucleotide Transformer in downsteam tasks.

| | EMP | TF | CPD | PD | SSP | EAP |
|---|---|---|---|---|---|---|
| Optimizer | | | AdamW | | | |
| Optimizer momentum | | | $\beta_1, \beta_2 = 0.9, 0.999$ | | | |
| Batch size | 32 | 32 | 32 | 32 | 32 | 64 |
| Training epoch | 100 | 10 | 10 | 5 | 10 | 10 |
| Learning rate | 3e-5 | 1e-4 | 1e-4 | 1e-4 | 1e-4 | 1e-4 |
| Weight decay | | | 0 | | | |

Table 8: Default hyperparameter settings for HyenaDNA in downsteam tasks

| | SSP | EMP | CPD&PD | EAP |
|---|---|---|---|---|
| Optimizer | | AdamW | | |
| Optimizer momentum | | $\beta_1, \beta_2 = 0.9, 0.999$ | | |
| Batch size | | 256 | | |
| Training epoch | | 100 | | |
| Learning rate | 6e-4 | 6e-4 | 7e-4 | 6e-4 |
| Weight decay | $0.20.0^7, 0.2^8$ | $0.0^{1,3,4}, 0.1, 0.2^5$ | 0.0 | 0.2 |
| Embed dropout | 0.1 | $0.0, 0.1^{1,3,5}, 0.2^2$ | 0.0 | 0.2 |
| Resid dropout | $0.1^7, 0.2^8$ | $0.0^6, 0.1, 0.2^5$ | 0.1 | 0.1 |
| Reverse complement aug. | false | false | true | false |

[1]H3, [2]H3K4me1, [3]H3K4me2, [4]H3K36me3, [5]H4, [6]H4ac, [7]splice site acceptor, [8]splice site donor

## A.8 Downstream Task Embedding

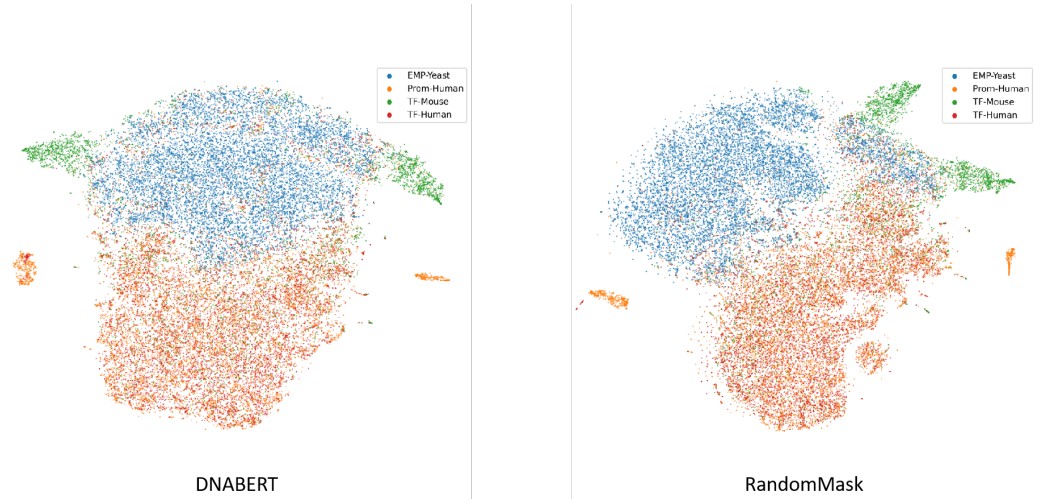

Figure 4: t-SNE of the downstream tasks' embeddings generated by DNABERT and DNABERT + Random-Mask.

DNABERT using RandomMask produce clearer boundaries, and the separation between different species is more obvious.