# OpenReview forum: "Rethinking the bert-like pretraining for dna sequences"
_ICLR.cc/2024/Conference — Submitted to ICLR 2024_

### Official Review · Reviewer_kyqC · 2023-10-24

**Soundness:** 2 fair
**Presentation:** 2 fair
**Contribution:** 2 fair
**Rating:** 6
**Confidence:** 3

**Summary:**

This paper discusses the growing trend of applying large-scale NLP-style pretraining to life sciences, particularly in DNA sequence analysis. It highlights the limitations of existing methods, the advantages of K-mer overlapping tokenization in downstream tasks, and the introduction of "RandomMask," a novel approach that significantly improves pretraining performance in life sciences applications, achieving remarkable results in epigenetic mark prediction.

**Strengths:**

Thorough analytical background on the method

**Weaknesses:**

Not much applicable to general machine learning, too specific in bioinformatics

- In the description of Table 2, what are MCC and PCC?  The reviewer is aware that they are later explained in Experiemnt section, but what they are and what they do need to be briefly explained in the description of Table 2 as well for the readers who are not in life sciences field.

- The authors say they progressively expand the masking boundary to prevent easy learning. Shouldn't it be the other way around? The reviewer believes that the masking boundary should progressively contract. What is the point of showing the shorcut first and complicating the learning? The model would already learn the shortcut if the masking boundary progressively expanded. An additional experiment on this needs to be conducted.

---

**Post Rebuttal**

Well addressed!

**Questions:**

Refer to Weaknesses

**Details Of Ethics Concerns:**

Refer to Weaknesses

---

> ### Author Response · Authors · 2023-11-21
>
> **Q1**: In the description of Table 2, what are MCC and PCC? The reviewer is aware that they are later explained in Experiemnt section, but what they are and what they do need to be briefly explained in the description of Table 2 as well for the readers who are not in life sciences field.
>
>
> **A1**: I have put the following content into the description of Table 2 or Supplementary Material.
>
> Matthews Correlation Coefficient (MCC) and Pearson Correlation Coefficient (PCC) are both statistical measures used to assess the relationship between variables. The maximum values of MCC and PCC are both 100%. The larger these two indicators are, the better the model performance is.
>
> 1. **Matthews Correlation Coefficient (MCC)**: MCC is primarily used in binary classification problems to measure the performance of binary classification models. The formula for MCC is:
>    $$
>    MCC = \frac{TP \times TN - FP \times FN}{\sqrt{(TP + FP)(TP + FN)(TN + FP)(TN + FN)}}
>    $$
>
>    where:
>    -  TP  = Number of True Positives
>    -  TN  = Number of True Negatives
>    -  FP  = Number of False Positives
>    -  FN  = Number of False Negatives
>
>    True Positives are instances where the model predicts the sample as positive, and it is indeed positive. True Negatives are instances where the model predicts the sample as negative, and it is indeed negative. False Positives are instances where the model predicts the sample as positive, but it is actually negative. False Negatives are instances where the model predicts the sample as negative, but it is actually positive.
>
> 2. **Pearson Correlation Coefficient (PCC)**: PCC is used to measure the strength and direction of the linear relationship between two continuous variables. Let $\hat{y_i}$ be the model's predicted value, and $y_i$ be the actual value of the sample. The formula for PCC is:
>   $$
>    PCC = \frac{\sum_{i=1}^n (\hat{y_i} - \bar{\hat{y}})(y_i - \bar{y})}{\sqrt{\sum_{i=1}^n (\hat{y_i} - \bar{\hat{y}})^2 \sum_{i=1}^n (y_i - \bar{y})^2}}
>   $$
>
>    where:
>    - $\hat{y_i}$ = Predicted value of the sample by the model
>    - $y_i$ = Actual value of the sample
>    - $\bar{\hat{y}}$ = Mean of the model's predicted values
>    - $\bar{y}$ = Mean of the actual values
>
>    EMP, TF-M, TF-H, PD, CPD, and SSP are classification tasks, so the evaluation metric for the model on these tasks is MCC. EAP is a regression task, therefore the evaluation metric for the model on this task is PCC.

---

> ### Author Response · Authors · 2023-11-21
>
> **Q2**: The authors say they progressively expand the masking boundary to prevent easy learning. Shouldn't it be the other way around? The reviewer believes that the masking boundary should progressively contract. What is the point of showing the shorcut first and complicating the learning? The model would already learn the shortcut if the masking boundary progressively expanded. An additional experiment on this needs to be conducted.
>
>
> **A2**:
>
> Curriculum learning was proposed  by Yoshua Bengio in 2009. It is a training strategy that mimics the human learning process by presenting the examples in an easy-to-difficult order. It has been shown to improve the performance and convergence stability of various Large language model.
>
> The paper in ACL2020 that explore curriculum learning for Large language Model, titled *Curriculum Learning for Natural Language Understanding*. This paper proposes a curriculum learning approach that reviews the training set in a crossed way to distinguish easy and hard examples, and arranges a curriculum for Large language Model on various NLU tasks. We wanted to increase the difficulty of the DNA sequence pre-training task and at the same time ensure the stability of the training, so we adopted a course learning solution.
>
>
>
> In the following table, the best performance results are represented by **boldface**, and the second best performance results are represented by <u>underline marks</u>. We tried consecutive mask6, 12, 14, 28 and 40 tokens during the entire pre-training process. Finally, it was found that this RandomMask from mask 6 to mask 14 achieved the best performance.
> | mask            | avg.  | EMP    | TF-M   | TF-H   | PD     | CPD    | SSP    | EAP    |
> |-----------------|-------|--------|--------|--------|--------|--------|--------|--------|
> | Mask 6 (DNABERT)| 70.73 | 51.81  | 60.40  | 64.10  | 93.05* | **71.88** | 85.44  | 68.43  |
> | Mask 12         | 71.98* | 52.96* | 67.12* | 67.05  | 92.01  | 71.05* | 86.62* | 67.03  |
> | Mask 14         | 71.80 | 51.44  | 66.86  | 66.60  | 92.59  | 70.67  | 85.31  | 69.16* |
> | Mask 28         | 70.94 | 50.95  | 65.94  | 67.75* | 91.03  | 70.70  | 83.33  | 66.85  |
> | Mask 40         | 69.94 | 49.93  | 66.19  | 67.54  | 90.21  | 70.01  | 82.35  | 63.37  |
> | RandonMask      | **74.85** | **65.83** | **67.96** | **69.73** | **93.40** | 70.24  | **87.20** | **69.56** |
>
> We further tested both options from easy to hard and from hard to easy. In the table below, we divide pre-training into two stages equally. Mask 6 to 12 means masking 6 tokens continuously in the first stage and masking 12 tokens continuously in the second stage. Mask 12 to 6 means masking 12 tokens continuously in the first stage and masking 6 tokens continuously in the second stage. It can be found that the former has better performance.
> | strategy    | avg.  | EMP   | TF-M | TF-H | PD    | CPD   | SSP  | EAP  |
> |-------------|-------|-------|------|------|-------|-------|------|------|
> | Mask 6 to 12| **73.06** | **59.22** | **67.83** | **66.37** | **92.81** | 71.51 | **86.20** | **67.38** |
> | Mask 12 to 6| 70.64 | 53.15 | 63.47 | 65.18 | 90.42 | **71.93** | 83.95 | 66.37 |

---

### Official Review · Reviewer_45WA · 2023-10-31

**Soundness:** 4 excellent
**Presentation:** 3 good
**Contribution:** 3 good
**Rating:** 6
**Confidence:** 3

**Summary:**

The author presented three observations by comparing the overlapping with non-overlapping tokenization method: (a) overlapping tokenization is always better than non-overlapping tokenization in the fine-tuning phase; (b) the loss overlapping tokenization method drops rapidly and result in a fast convergence of embedding space; (c) focused attention has been observed in the overlapping tokenization method. Both (b) and (c) indicate the overfitting problem in overlapping tokenization. The author proposed a novel RandomMask method to mitigate the overfitting issue by gradually increasing the complexity of the task. Based on the author’s experiment, the RandomMask+BERT outperformed the other benchmarks algorithms.

**Strengths:**

[Originality and Significance] The author presented two original findings. One is that the overlapping tokenization always performs better in the fine tuning stage, regardless of the tokenization method in the pre-training. This is different from the conventional wisdom and could provide the insights for many related research and applications. The second contribution is that gradually increasing the complexity during the training could mitigate the overfitting issue and achieve better performance by combining fast convergence and generalizability. This method could inspire other researchers to consider similar techniques to balance the convergence and generalizability. The performance of the RandomMask algorithm is better than the precedent algorithms in different tasks, which is a significant result for the DNA sequence analysis.

[Organization] The paper is presented in a well-organized way, from the observations to the new algorithms. Both the them bring new knowledge to the field.

**Weaknesses:**

[Clarity] The experiment and result section is relatively short. Some more clarifications would be helpful. For example, (a) when describing the baseline, the author mentioned “All models are trained on human genome and fine-tuned on the benchmark datasets with identical settings.” The “identical” setting does not bring clarity on how the mask is generated. Does this mean all the algorithms are trained and fine tuned with 15% token masked? This setting is slightly different than the original setting in the “Nucleotide Transformer” paper. (b) “Finetuning” section discussed the dataset in a detailed way; however, the details of the algorithm setup in the fine tuning is not discussed.

[Quality] The value of RandomMark (or training with gradually increased difficulty) can be better verified through experiments. For example, the author presented the result using the training with 5 phases. It would be great if the author could compare the result with different numbers of phases: one phase with maximum difficulty, two phases with two different difficulties, and etc.

**Questions:**

Q1: Algorithm 1 is not very clear to me. For example, if the r <= p at position i, [i − m/2 + 1, i + m/2] will be masked. The following nucleotide should be i+1, or I+m/2+1? If the next one is i+1, the mask range could be overlapping and can grow to be very long.

---

> ### Author Response · Authors · 2023-11-21
>
> **[Clarity]**: The experiment and result section is relatively short. Some more clarifications would be helpful. For example, (a) when describing the baseline, the author mentioned “All models are trained on human genome and fine-tuned on the benchmark datasets with identical settings.” The “identical” setting does not bring clarity on how the mask is generated. Does this mean all the algorithms are trained and fine tuned with 15% token masked? This setting is slightly different than the original setting in the “Nucleotide Transformer” paper. (b) “Finetuning” section discussed the dataset in a detailed way; however, the details of the algorithm setup in the fine tuning is not discussed.
>
> **A to Clarity**: We will additionally add detailed settings to the revised paper and open source our code.
>
> a) "Pretraining": For Nucleotide Transformer, hyenaDNA, and DNABERT, we used the open source pre-training weights from huggingface. When we pretrain DNABERT using RandomMask (DNABERT+RandomMask), we just modified the strategy for masking with RandomMask. The Other settings, such as the model structure, training steps, and learning rate, remained consistent with the settings of DNABERT's paper. We have made changes to the introduction of pretraining in the experimental section.
>
> b) "Finetuning": In the fine-tuning stage, the fine-tuning parameters for NT are detailed in the A.7 of supplementary material's final section. The fine-tuning parameters for DNABERT and DNABERT+RandomMask are identical and their specific values can be found in the A.7 of supplementary materials. HyenaDNA used the fine-tuning parameters, being consistent with HyenaDNA's paper, can also be found in the A.7 of supplementary materials.
>
> **[Quality]** The value of RandomMark (or training with gradually increased difficulty) can be better verified through experiments. For example, the author presented the result using the training with 5 phases. It would be great if the author could compare the result with different numbers of phases: one phase with maximum difficulty, two phases with two different difficulties, and etc.
>
> **A to Quality**:
>
> (1) Two phases.
>
> In the following table, we tested both options from easy to hard and from hard to easy. In the table below, we divide pre-training into two stages equally. Mask 6 to 12 means masking 6 tokens continuously in the first stage and masking 12 tokens continuously in the second stage. Mask 12 to 6 means masking 12 tokens continuously in the first stage and masking 6 tokens continuously in the second stage.
>
> In the following table, the best performance results are represented by **boldface**
> | strategy    | avg.  | EMP   | TF-M | TF-H | PD    | CPD   | SSP  | EAP  |
> |-------------|-------|-------|------|------|-------|-------|------|------|
> | Mask 6 to 12| **73.06** | **59.22** | **67.83** | **66.37** | **92.81** | 71.51 | **86.20** | **67.38** |
> | Mask 12 to 6| 70.64 | 53.15 | 63.47 | 65.18 | 90.42 | **71.93** | 83.95 | 66.37 |
>
> It can be found that the easy to hard strategy has better performance. This verifies the rationality of the course learning strategy from easy to difficult in BERT-like DNA pre-training.
>
>
> (2) Compare one and two phase.
>
> We tried consecutive mask 6, 12, 14, 28 and 40 tokens during the entire pre-training process. DNABERT + RM (RandomMask) means use RandomMask to pretrain DNABERT.
>
> In the following table, the best performance results are represented by **boldface**, and the second best performance results are represented by *.
> | mask            | avg.  | EMP    | TF-M   | TF-H   | PD     | CPD    | SSP    | EAP    |
> |-----------------|-------|--------|--------|--------|--------|--------|--------|--------|
> | Mask 6 (DNABERT)| 70.73 | 51.81 | 60.40  | 64.10  | **93.05** | **71.88** | 85.44  | 68.43  |
> | Mask 12         | 71.98 | 52.96 | 67.12 | 67.05 | 92.01  | 71.05* | 86.62* | 67.03  |
> | Mask 14         | 71.80 | 51.44  | 66.86 | 66.60  | 92.59*  | 70.67  | 85.31  | 69.16* |
> | Mask 28         | 70.94 | 50.95  | 65.94  | 67.75*  | 91.03  | 70.70  | 83.33  | 66.85  |
> | Mask 40         | 69.94 | 49.93  | 66.19  | 67.54  | 90.21  | 70.01  | 82.35  | 63.37  |
> | Mask 6 to 12| 73.06* | 59.22* | 67.83* | 66.37 | 92.81* | 71.51 | 86.20 | 67.38 |
> | DNABERT + RM       | **74.75** | **65.83**| **67.96** | **69.73** |92.74| 70.24 | **87.20**| **69.56**|
>
> From the table above, we can see that the length of the continuous mask is not longer, the better. The optimal performance of various tasks is concentrated around mask 12.
>
> From the above results, we designed a RandomMask strategy that gradually increases the task difficulty in five stages from 8 to 14.

---

> ### Author Response · Authors · 2023-11-22
>
> **Q1**: Algorithm 1 is not very clear to me. For example, if the r <= p at position i, [i − m/2 + 1, i + m/2] will be masked. The following nucleotide should be i+1, or I+m/2+1? If the next one is i+1, the mask range could be overlapping and can grow to be very long.
>
> **A1**: If the r <= p at position i, [i − m/2 + 1, i + m/2] will be masked. the following tokens should be i+1. In fact, the mask range can grow to be very long. We conducted the following experiments when selecting strategies.
>
> In the following table, the best performance results are represented by **boldface**, and the second best performance results are represented by *. We tried consecutive mask 6, 12, 14, 28 and 40 tokens during the entire pre-training process. Finally, it was found that this RandomMask from mask 6 to mask 14 achieved the best performance.
> | mask            | avg.  | EMP    | TF-M   | TF-H   | PD     | CPD    | SSP    | EAP    |
> |-----------------|-------|--------|--------|--------|--------|--------|--------|--------|
> | Mask 6 (DNABERT)| 70.73 | 51.81  | 60.40  | 64.10  | 93.05* | **71.88** | 85.44  | 68.43  |
> | Mask 12         | 71.98* | 52.96* | 67.12* | 67.05  | 92.01  | 71.05* | 86.62* | 67.03  |
> | Mask 14         | 71.80 | 51.44  | 66.86  | 66.60  | 92.59  | 70.67  | 85.31  | 69.16* |
> | Mask 28         | 70.94 | 50.95  | 65.94  | 67.75* | 91.03  | 70.70  | 83.33  | 66.85  |
> | Mask 40         | 69.94 | 49.93  | 66.19  | 67.54  | 90.21  | 70.01  | 82.35  | 63.37  |
> | RandonMask      | **74.85** | **65.83** | **67.96** | **69.73** | **93.40** | 70.24  | **87.20** | **69.56** |
>
> When setting a continuous mask of 14 and mask rate p=2.5%, approximately 25% of the tokens in the entire sequence are covered. In other words, in this case, the maximum number of tokens that can be continuously covered is not too long, no more than a quarter of the sequence length.
>
> It is not that the longer the continuous mask, the better. When the number of consecutive masks is greater than 14, it is possible that the mask may have too long consecutive tokens due to the overlap you mentioned, further leading to a decline in model's performance.

---

> > ### Comment · Reviewer_45WA · 2023-11-22
> > **Adjusted rating**
> >
> > Thank you for the followup experiment. I really appreciate the soundness of your argument, which was accordingly updated.

---

> > > ### Author Response · Authors · 2023-11-23
> > >
> > > It's great to hear that you appreciate our work. And thanks very much for your valuable suggestions. If you have any further questions, feel free to contact us.

---

### Official Review · Reviewer_g7iT · 2023-11-02

**Soundness:** 3 good
**Presentation:** 2 fair
**Contribution:** 3 good
**Rating:** 3
**Confidence:** 3

**Summary:**

#### post-discussion
---
After the discussion I decide to keep my rating.

Overall, I think the empirical findings and the proposed RandomMask could contribute to the DNA representation learning community. However, my main concern is the presentation of this paper.

1. Current version of paper could be misleading in terms of the topic. Tokenization/masking strategies are important but not everything. The authors justified that the "primary influence on BERT-like pre-training are tokenization and masking", which does not make sense to me. We have other problems like the training objectives, positional encoding and sequence segmentation/construction, etc.
2. The three observations and the proposed RandomMask are both interesting. But the Section3 (observations) and Section4 (the method) are not connected well. It would be hard for the readers to understand why/how RandomMask can help with the issues discussed in Section3.

I didn't find these issues addressed in the updated manuscript. In fact, I believe a re-organization of the paper is needed (e.g., make the observation more concise and the sections more integrated). Reviewer SHyo and 45WA also mentioned the clarification issues. We will not publish the paper with all the rebuttal materials together, so the readability of the manuscript itself is important.

#### previous review
---
The authors of the paper initially conduct empirical experiments and analysis on the use of overlapping tokenization in BERT-like pretraining for DNA sequences. Their observations reveal that: 1) overlapping tokenization consistently enhances fine-tuning performance; 2) models trained with overlapping converge more rapidly; 3) overlapping can result in sparse attention within intermediate layers.

The authors claim that above observations demonstrate the limitation of existing overlapping tokenization. Subsequently, the authors introduce a dynamic overlapping strategy, referred to as RandomMask, for the pretraining of DNA sequences. Experimental evidence from a range of downstream tasks suggests that RandomMask consistently improves performance.

**Strengths:**

* Extensive empirical results and analysis, providing some findings about overlapping strategy in DNA tokenization, could benefit the community.
* The proposed method RandomMask achieves SOTA on various downstream tasks.
* The proposed RandomMask is effective but simple. It could be easy to be re-implemented and deployed for further research.

**Weaknesses:**

While this paper provides extensive empirical results and quantitively demonstrates the effectiveness of RandomMask, there are several areas where it could be further enhanced. My main concerns are as follows:

1. The authors might consider refining the focus of this work. The true contribution appears to be the improvement of the overlapping strategy tokenization for DNA pretraining, which diverges from the broader theme of "rethinking the pretraining for DNA sequence."

1. The motivation behind the study is somewhat unclear. Although the authors identify three potential challenges -- rapid convergence, the risk of under-training, and the potential for sparse attention -- they do not adequately explain how RandomMask addresses or mitigates these issues.

1. There is a lack of experimental analysis supporting the source of the observed improvements, which is crucial for substantiating the paper's main claims. For example, besides the quantative improvements, does the rapid convergence and under-training still exist after applying RandomMask?

1. The comparison in Observation 1 does not seem to be an apples-to-apples comparison. Overlapping represents more patterns and creates longer sequences for the same DNA length. It would be beneficial to understand if the conclusion holds for different lengths of k-mer.

1.  The paper's presentation could be improved in several ways:
    1. The introduction is somewhat verbose, indirectly causing the first two weaknesses and making the paper hard to read.
    1. Placing Figure 1 and Table 1 on page 1 would improve readability, given that the main content describing Figure 1 and Table 1 is in the first page.
    1. The separate table on the left in Table 2 appears to be redundant.
    1. The experimental settings in Section 3 lack detailed descriptions, potentially making reproduction difficult and potentially misleading.
    1. A thorough proofreading could enhance the clarity of writing and word choice.

1. It would be beneficial to further explore whether sparse attention is indeed a problem for DNA sequence representation. Sometimes, sparse attention can improve generalization [1]. This might depend on different sub-sequences and the various functions of different layers when modeling cross-attention. I would appreciate further elaboration on the limitations of sparse attention in DNA sequence representation.

I would appreciate the explanation and further evidence to address these concerns.

[1] Correia, et al. "Adaptively Sparse Transformers." Conference on Empirical Methods in Natural Language Processing (2019).

**Questions:**

See Weakness section.

---

> ### Author Response · Authors · 2023-11-21
>
> **Q1**: The authors might consider refining the focus of this work. The true contribution appears to be the improvement of the overlapping strategy tokenization for DNA pretraining, which diverges from the broader theme of "rethinking the pretraining for DNA sequence."
>
> **A1**: We named our paper as "Rethinking the BERT-like Pretraining for DNA Sequences" based on the following two points:
>
> 1) The primary influences on BERT-like pretraining methods are the **tokenizer** and **mask**. Our paper discusses both of these key factors: 1) We discuss the impacts of DNA **tokenizer** in both pretraining and fine-tuning stages, as shown in Figure 1, Figure 2, Table 2 of manuscript and Table 1 of Supplementary Material. In Figure 1 of manuscript, we show the comparisons between commonly used BERT-like DNA tokenizer and NLP tokenizer. In Figure 2 of manuscript, we show the impact of overlapping and non-overlapping tokenizer on the pre-training process and the underlying representation of the model. In Table 2 of manuscript, we show the influence of overlapping and non-overlapping tokenizer on the fine-tuning stage. In Table 1 of the supplementary material, we also compare the performance of 6mer with BPE, with the support of RandomMask, the performance of 6mer is much higher than the most advanced BPE model. 2) We investigate the methods of **mask** in BERT-like pretraining. In Figure 1, we compare different mask approaches in DNA and NLP. In Table 4 of the supplementary material, we discussed the impact of the number of consecutive mask tokens on performance. In Table 5 of the supplementary material, we discussed pre-training mask task from easy to difficult and from difficult to easy. Based on these ablation experiments, we found the optimal setting strategy for RandomMask.
> 3) Our contribution includes **three experimental observations** related to BERT-like DNA pretraining and **a simple yet effective method**. Reviewer SHyo acknowledged the inspirational significance of these progressive observations, while Reviewer 45WA affirmed that our novel insights differ from traditional views, offering valuable insights for numerous related studies and applications. Based on these observations, our article introduces a simple yet effective method called RandomMask. Therefore, our contribution is not just in optimizing the tokenizer, but in providing three new observations regarding BERT-like DNA pretraining and presenting an effective method based on these insights.

---

> ### Author Response · Authors · 2023-11-21
>
> **Q2 and Q3**：The motivation behind the study is somewhat unclear. Although the authors identify three potential challenges -- rapid convergence, the risk of under-training, and the potential for sparse attention -- they do not adequately explain how RandomMask addresses or mitigates these issues. There is a lack of experimental analysis supporting the source of the observed improvements, which is crucial for substantiating the paper's main claims. For example, besides the quantative improvements, does the rapid convergence and under-training still exist after applying RandomMask?
>
> **A2 and A3**: This study gives a comprehensive understanding and a specifically tailored approach for bert-like DNA pretraining. Based on a series of exploratory experiments, three insightful observations are given, followed by a simple-but-effective method RandomMask for solving the rapid convergence and under-training problems. Below, we give more experimental analysis supporting the observed improvements of RandomMask:
>
> 1) In Figure 3 of Supplementary Material, we can see that DNABERT's loss will quickly decrease to an extremely low value. If RandomMask is used, the loss will increase at the start of each stage, giving it enough space to decrease, so we will see a decrease at each stage in the loss curve with RandomMask. Reviewer 45WA affirmed this point. He emphasized that we enhanced the generalization of the model by increasing the difficulty of the task and reducing the overfitting problem caused by rapid convergence that gradually increasing the complexity during the training could mitigate the overfitting issue and achieve better performance by combining fast convergence and generalizability.
> 2) In the Figure 1 of Supplementary Material, we can see that the middle layer only pays attention to the [CLS] token, while the first and last layers capture a variety of features. Further, in Figure 2 of Supplementary Material, we see that all tokens in the DNABERT's middle layer pay excessive attention to [CLS]. Therefore the middle layer is not sufficiently trained to capture sequence features. However, as showed in Figure 2 of Supplementary Material, if pre-training with RandomMask, these layers of the model are activated. The attention weights of these layers learn to pay attention to other tokens, and the attention patterns are more diverse. Diverse attention weights indicate that these layers are mroe adequately trained to capture sequence features as the first and last layers do.
> 3) What RandomMask wants to solve is the fixed attention weight pattern of the middle layers in DNABERT. For sparse attention, different tokens and different layers usually show diverse sparse patterns. However, in Figure 1 and 2 of Supplementary Material, we see that the different layers in the middle of DNABERT and all tokens show the same attention weight pattern, that is, concentrated attention [CLS]. Thus, the problem we're trying to solve and sparse attention are two different things. In Figure 2 of Supplementary Material, we see that adding RandomMask will diversify the attention weight pattern of layers in the middle of DNABERT.

---

> ### Author Response · Authors · 2023-11-21
>
> **Q4**: The comparison in Observation 1 does not seem to be an apples-to-apples comparison. Overlapping represents more patterns and creates longer sequences for the same DNA length. It would be beneficial to understand if the conclusion holds for different lengths of k-mer.
>
> **A4**: Follow the reviewer's suggestions, we investigate the effect of sequence length. The following table is an expansion of Table 2 in the paper. The row named Same-lenth is to copy the Non-overlapping tokens five times to obtain the same sequence length as Overlapping.
> (i.e. Non-overlapping: token1 token2; Same-lenth: token1 token2 token1 token2 token1 token2 token1 token2 token1 token2 token1 token2.)
>
> The best performance results are represented by **boldface**, and the second best performance results are represented by *.
> | NT              |    avg. |   EMP |   TF-M |   TF-H |    PD |   CPD |   SSP |   EAP |
> |----------------|--------|------|-------|-------|------|------|------|------|
> | Non-overlapping | 58.68 | 45.37* | 39.81 | 55.25 | 88.43* | 62.56 | 80.39 | 38.94* |
> | Same-lenth      | 60.42* | 44.88 | 47.59* | 60.57* | 86.96 | 63.98* | 80.96* | 37.98 |
> | Overlapping     | **68.69** | **46.47** | **61.99** | **63.95** | **90.88** | **68.55** | **84.34** | **64.67** |
>
> | DNABERT         |    avg. |   EMP |   TF_M |   TF_H |    PD |   CPD |   SSP |   EAP |
> |----------------|--------|------|-------|-------|------|------|------|------|
> | Non-overlapping | 60.24*   | 43.65* | 34.87  | 54.50*  | 87.62* | 65.82* | 79.91 | 55.31* |
> | Same-lenth      | 57.66 | 42.98 | 38.60*  | 53.27  | 85.33 | 64.09 | 80.76* | 38.62 |
> | Overlapping     | **70.49**   | **51.81** | **59.60**  | **63.55**  | **91.76** | **72.84** | **85.44** | **68.43** |
>
> 1) An interesting phenomenon. In NT that uses non-overlapping 6mer for pre-training, stretching the sequence length will indeed produce obvious gains in TF-M, TF-H and CPD. Combined with Table 1 in the paper, the common feature of these three tasks is that the DNA sequence length is short. The DNA sequence lengths of EMP, PD, SSP and EAP are 500, 300, 400 and 250 nucleotides respectively. However, the DNA sequence lengths of TF-M, TF-H and CPD are 100, 100 and 70 nucleotides respectively, and these are shorter than others.
>
> 2) But in general, the overlapping tokenizer to obtain more diverse tokens get better performance than simply lengthening the sequence length (Same-lenth) both of overlapping pretraining (DNABERT) model and non-overlapping pretraining model (NT).

---

> ### Author Response · Authors · 2023-11-21
>
> **Q5**: The paper's presentation could be improved in several ways:
>
> 1) The introduction is somewhat verbose, indirectly causing the first two weaknesses and making the paper hard to read.
> 2) Placing Figure 1 and Table 1 on page 1 would improve readability, given that the main content describing Figure 1 and Table 1 is in the first page.
> 3) The separate table on the left in Table 2 appears to be redundant.
> 4) The experimental settings in Section 3 lack detailed descriptions, potentially making reproduction difficult and potentially misleading.
> 5) A thorough proofreading could enhance the clarity of writing and word choice.
>
> **A5**: Thanks for your valuable suggestions, we have considered all the suggestions and improved the paper's presentation as much as we can. Besides, we will ask senior researchers who have published more than ten top conference papers to revise the manuscript to ensure that the article is concise and easy to read.
>
>
>
>
> **Q6**: It would be beneficial to further explore whether sparse attention is indeed a problem for DNA sequence representation. Sometimes, sparse attention can improve generalization [1]. This might depend on different sub-sequences and the various functions of different layers when modeling cross-attention. I would appreciate further elaboration on the limitations of sparse attention in DNA sequence representation.
>
> [1] Correia, et al. "Adaptively Sparse Transformers." Conference on Empirical Methods in Natural Language Processing (2019).
>
> **A6**: In A.5 of the Supplementary Material, we show that DNABERT faces the problem of fixed attention weight patterns rather than the problem of sparse attention weights. Specifically, for DNABERT-6mer with open source weights, the pattern of attention weights in intermediate layers such as layers 4 and 5 is very fixed (only focusing on [CLS] token). Such results reveal that the original DNABERT is not sufficiently trained. Thus, we propsoe RandomMask to alleviate this problem.
>
> For sparse attention, we find some discussions in GENA-LM [2]. The following table results are from GENA-LM. As we can see, sparse attention does show significant gains in some down-stream tasks like enhancer-development and polyadenylation site prediction, but other tasks performed essentially the same or worse.
>
> | Model          |    avg. |   prom |   deepsea-DHS |   deepsea-TF |   deepsea-HM |   enhancer-HouseKeeping |   enhancer-Development |   polyadenylation site prediction |
> |---------------|--------|-------|--------------|-------------|-------------|------------------------|-----------------------|----------------------------------|
> | Sequence length |--- | 300  | 1000        | 1000       | 1000       | 249                  | 249                  | 450                             |
> | DNABERT        | 82.50 | 74.56| **91.83**   | **95.69**  | **85.17**  | 75.7                 | 64.6                 | 89.92                           |
> | DNABERT-sparse | **82.55** | **74.57** | 90.73   | 94.96      | 84.72      | **75.9**             | **65.7**             | **91.26**                       |
>
> [2] Fishman, Veniamin, et al. "GENA-LM: A Family of Open-Source Foundational Models for Long DNA Sequences." bioRxiv (2023): 2023-06.

---

> > ### Comment · Reviewer_g7iT · 2023-11-23
> >
> > Thanks for the authors' effort and detailed clarification! Some of my concerns have been addressed (especially the experimental part) but others about the presentation are not yet.
> >
> > I have a quick question. "The primary influences on BERT-like pretraining methods are the tokenizer and mask." Could you please provide reference to support this claim? It's not very direct to me.
> >
> > Overall, my main concern at this moment is presentation of this paper. It has clear technical contribution (I gave 3 for contribution) however readers might be not very easy to obtain insights from it. Based on the clarification, I will increase the presentation score from 1 to 2. However, I will make the final decision on the overall rating after reading other rebuttal contents and discussing with other reviewers.

---

### Official Review · Reviewer_SHyo · 2023-11-03

**Soundness:** 2 fair
**Presentation:** 3 good
**Contribution:** 3 good
**Rating:** 6
**Confidence:** 4

**Summary:**

The paper conducted an extensive study on using BERT-like models for pretraining with DNA sequences. Their experiments showed that using a tokenization approach that overlaps K-mers gives better results during the fine-tuning stage, regardless of whether the pretraining involved overlapping or not. However, they found that the commonly used method of overlapping tokenization during the pretraining phase caused the model to converge too quickly, which resulted in inadequate training.

To tackle this issue, the authors introduced the Random-Mask method. This method involves pretraining with dynamically changing the boundaries of the masked sections, which pushes the model to assimilate richer knowledge. They observed that when they expanded the mask boundaries during different training phases, there was a notable increase in the loss value. This increase in loss suggests that the model encounters new challenges and continues to learn, as evidenced by a downward trend in the training curve for each phase where the mask boundary is expanded.

They tested their approach on a total of 28 datasets spanning 7 downstream tasks. Across these tasks, we consistently achieved top-tier performance.

**Strengths:**

The paper is well-written and presents its findings in a clear and logical manner, effectively explaining all observations and results. I especially like how it points out important observations step by step until it introduces the new technique. The graphs showing how the model's errors changed during training, attention maps and t-sne plots that help visualize the data made it easier to get what the paper is saying.

**Weaknesses:**

The evaluation section is lacking in clarity. It would be helpful to answer the questions listed below and help readers understand how RandomMask overall improves the performance of downstream tasks.

**Questions:**

(1) How does the RandomMask method compare with alternate tokenization approaches such as BPE (proposed in DNABERT-2)?
(2) How does the RandomMask improve the internal representations - Can you see visible differences in embedding representations for downstream tasks (e.g Biotype Embeddings shown in HyenaDNA)
(3) How does RandomMask compare on the benchmark datasets listed in  HyenaDNA?

---

> ### Author Response · Authors · 2023-11-21
>
> **Q1**: How does the RandomMask method compare with alternate tokenization approaches such as BPE (proposed in DNABERT-2)?
>
> **A1**: Follow the reviewer's suggestion, we conduct comprehensive experiments to compare our RandomMask (RM) method with BPE and DNABERT-2. Here we name the open source DNABERT-2 model as DNABERT2-BPE to distinguish it from the DNABERT2-6mer that we trained.
>
> (1) Compare DNABERT+RM and DNABERT2-BPE
>
> In the table below, we compare our DNABERT + RM (RandomMask), with open source DNABERT and DNABERT2-BPE, the best performance results are represented by **boldface**, and the second best performance results are represented by *.
> | Model              | Avg.  | EMP   | TF-M  | TF-H  | PD   | CPD   | SSP   | EAP   |
> |--------------------|-------|-------|-------|-------|------|-------|-------|-------|
> | DNABERT            | 70.36 | 51.81 | 60.40 | 64.10 |90.48*| **71.88** | 85.44* | 68.43* |
> | DNABERT + RM       | **74.75** | **65.83** | 67.96* | 69.73* |**92.74**| 70.24 | **87.20** | **69.56** |
> | DNABERT2-BPE       | 73.40* | 64.47* | **68.00** | **70.11** | 87.91 | 70.53* | 84.99 | 67.79 |
>
> As you can see from the table below, the performance of our pre-trained DNABERT+RM is slightly better than DNABERT2-BPE.
>
> (2) Compare DNABERT2+6mer and DNABERT2-BPE
>
> In addition to using the BPE tokenizer method, DNABERT-2 also changed the model architecture, adopted a new activation function, and the pre-training data are larger multi-species DNA data.
>
> In order to have a fair comparison, we trained DNABERT2-6mer and DNABERT2-6mer + RM (RandomMask) using 6mer tokenizer under the same architecture, pre-training data and pre-training hyperparameters of DNABERT-2's paper. The results are shown in the following table, the best performance results are represented by **boldface**, and the second best performance results are represented by *.
> | Model              | Avg.  | EMP   | TF-M  | TF-H  | PD   | CPD   | SSP   | EAP   |
> |--------------------|-------|-------|-------|-------|------|-------|-------|-------|
> | DNABERT2-BPE       | 73.40 | 64.47 | 68.00*| 70.11*| 87.91 | 70.53*| 84.99 | 67.79 |
> | DNABERT2-6mer      | 69.41 | 48.22 | 65.32 | 64.87 |88.55 | 68.19 |  84.36 | 66.37 |
> | DNABERT2-6mer + RM | **76.91** | **68.16** | **76.28** | **70.99** | 90.68*| **72.97** | **88.91** | **70.41** |
>
> In the table above, we see the results in the first and second rows and find that if we just replace the BPE with 6mer, the performance of the model will decrease. The third line is the model performance after using RandomMask. It can be seen that the performance of the 6mer model has been greatly improved after using RandomMask, far exceeding DNABERT2-BPE.
>
> **Q2**: How does the RandomMask improve the internal representations - Can you see visible differences in embedding representations for downstream tasks (e.g Biotype Embeddings shown in HyenaDNA)
>
> **A2**: Since HyenaDNA does not open source the code and biotype dataset for this task. Therefore we tried it on our dataset.
>
> Following the motivation of biotype, we selected EMP-Yeast, TF-Human, TF-Mouse and Prom-Human, covering three species and three functional originals. We used DNABERT and DNABERT + RandomMask to generate embeddings for these datasets, and performed t-SNE for visualization. We put the visualization results in A.8 of the Supplementary Material.
>
> We can get the following conclusion. Compared with the original DNABERT, the model pre-trained using RandomMask can produce clearer boundaries and the separation between different species is more obvious.

---

> ### Author Response · Authors · 2023-11-21
>
> **Q3**: How does RandomMask compare on the benchmark datasets listed in HyenaDNA?
>
> **A3**: Follow the reviewer’s suggestion, we compare RandomMask on the benchmark datasets listed in HyenaDNA:
>
> (1) 18 of the 25 downstream task datasets open sourced by HyenaDNA are already included in the GUE benchmark we used (including EMP, TF-M, PD, CPD, SSP and EAP). These results are showed in the Table 3 of the manuscript. The results of the remaining 7 datasets are as follows.
>
> (2) In the table below, the first row shows the results of CNN on the 7 datasets. The second row is the result of two layers of Transformers. The third row is the result of two layers of Hyena. These three rows of results are all from the original paper of HyenaDNA. We also test open source DNABERT and our DNABERT + RM (RandomMask), and show the results in the fourth and fifth rows respectively. Because the results in the original HyenaDNA's paper are accurate to one decimal place, our test results in this table are also kept to one decimal place. The best performance results are represented by **boldface**, and the second best performance results are represented by *.
> |                       | Avg.       | Coding vs Intergenomic | Human vs Worm | Human Enhancers Cohn | Human Enhancers Ensembl | Human OCR Ensembl | Human Regulatory | Human Nontata Promoters |
> |------------------------|--------------|------------------------|---------------|----------------------|-------------------------|-------------------|------------------|------------------------|
> | CNN                   | 80.7         | 87.6                   | 93.0          | 69.5                 | 68.9                    | 68.0              | <u>93.3</u>     | 84.6                   |
> | Transformer           | 84.4        | 88.8                   | 95.6          | 70.5                 | 83.5                    | 73.0              | 91.5            | 87.7                   |
> | HyenaDNA              | 88.3* | 87.6                   | 96.5          | 73.8                 | 89.2*             | 80.9              | **93.8**        | **96.6**               |
> | DNABETR              | 87.8        | 93.2*           | **97.0**      | 74.3*          | 89.0                    | 81.0*       | 88.5            | 91.7                   |
> | DNABERT + RM            | **89.4**    | **94.5**               | 96.9*   | **76.6**             | **91.3**                | **82.7**          | 90.3            | 93.4*            |
>
> Comparing the last two rows, we can find that except Human vs Worm, DNABERT+RM is always better than DNABERT. And the average performance of DNABERT+RM far exceeds DNABERT. This proves the superiority of RandomMask. Compared to other methods, DNABERT+RM achieves the best performance on four of the seven datasets. DNABERT+RM achieve second-best performance on two of the three datasets where performance was not the best. And DNABERT+RM are the best in terms of average performance across seven datasets.

---

### Meta-Review · Area_Chair_nb8A · 2023-12-06

**Metareview:**

This paper presents an investigation into pretraining BERT-like models using DNA sequences. Notably, the study explores the benefits of overlapping K-mer tokenization and introduces a novel pretraining method, termed RandomMask, which dynamically changes the masked regions to enhance training. The paper's sound empirical analysis is evident through extensive experiments across various datasets and tasks, offering insights into BERT-like DNA pretraining.

Strengths:
- The paper offers a clear explanation of its observations and results, backed by a logical progression of insights culminating in the RandomMask technique.
- Methodological soundness is evident, establishing the RandomMask approach as an effective tool for addressing the shortcomings of commonly used overlapping tokenization methods.
- The structured presentation and organization of the paper enable a smooth narrative, guiding readers from empirical observations to the conception of the new algorithm.

Weaknesses:
- The evaluation section lacks clarity, with the reviewer requesting further comparisons and deeper analyses to elucidate RandomMask's internal representation improvements and its impact on downstream tasks. The experiment and result section is relatively short.
- There is a notable specificity to the field of bioinformatics, with the paper's content being less applicable to general machine learning audiences.
- As mentioned by Reviewer g7iT, the authors might consider refining the focus of this work.  The true contribution appears to be the improvement of the overlapping strategy tokenization for DNA pretraining, which diverges from the broader theme of "rethinking the pretraining for DNA sequence." The response from authors is not convincing. This makes the paper appear quite ad-hoc.
- The main contributions of this paper seem lie in comprehensive experiments and analysis, but it lacks impressive analysis and conclusions (most analysis are quite as expected), thus leading to relatively less new findings.

**Justification For Why Not Higher Score:**

This is a bordeline paper, but as a rethinking paper, it lacks clarity in the evaluation and experimental analysis, and it also lacks impressive analysis and conclusions (most analysis are quite as expected), thus leading to relatively less new findings.

**Justification For Why Not Lower Score:**

N/A

---

### Decision · Program_Chairs · 2024-01-16

Reject